# A randomized trial to evaluate a complex, co-created, culture-sensitive intervention to promote healthy lifestyles and compliance to therapy in immigrants with type 2 diabetes: A protocol of a multicenter Italian study

**Laura Bonvicini[1], Francesco Venturelli[1]\*, Francesca Bononi[1], Giulietta Luul Balestra[2], Giusy Iorio[2], Luca Ghirotto[2], Alessio Petrelli[3], Silvia Pierconti[3], Giovanna Laurendi[3], Maria Perticone[4], Alessio Pellegrino[5,6], Maria Boddi[5,6], Pietro Amedeo Modesti[5,6], Paolo Giorgi Rossi[1], DIABETHIC Working Group[¶]**

1 Epidemiology Unit, Azienda USL-IRCCS di Reggio Emilia, Reggio Emilia, Italy, 2 Qualitative Research Unit, Azienda USL-IRCCS di Reggio Emilia, Reggio Emilia, Italy, 3 Epidemiology Unit, National Institute for Health, Migration and Poverty (INMP), Rome, Italy, 4 Department of Medical and Surgical Sciences, University Magna Graecia of Catanzaro, Catanzaro, Italy, 5 Medicina dello Sport e dell'Esercizio Fisico, Azienda Ospedaliero-Universitaria Careggi, Florence, Italy, 6 Department of Medicina Sperimentale e Clinica, University of Florence, Florence, Italy

¶ Membership of the DIABETHIC Working Group is listed in the Acknowledgments.
\* francesco.venturelli@ausl.re.it

## Abstract

### Introduction

The active involvement of end users may overcome socio-economic, cultural and context-related barriers that may reduce health promotion effectiveness in type 2 diabetes control and prevention. The "Cardio-metabolic diseases in immigrants and ethnic minorities: from epidemiology to new prevention strategies" (DIABETHIC) project funded by the European Union through the Italian Ministry of Health includes a multicentre randomised controlled trial (RCT) aimed to assess the effectiveness of a co-created health promotion intervention for immigrants affected by type 2 diabetes. This protocol describes the co-creation process and methodological challenges in evaluating co-created health promotion interventions.

### Methods and analysis

Between November 2023 and July 2024, four Italian primary care centres will recruit 200 immigrants with type 2 diabetes that will be randomised to usual health promotion practice or to the experimental health promotion intervention developed through a participatory process. Endpoints are changes in glycated haemoglobin, Body Mass Index, diet, physical activity and therapeutic adherence at 12 months after recruitment. Qualitative research experts supported the participatory process at local and national levels. According to available evidence and recommendations, the participatory process focused on the

**Data availability statement:** No datasets were generated or analysed during the current study.

**Funding:** Funded by the European Union – Next Generation EU – PNRR M6C2 - Investimento 2.1 Valorizzazione e potenziamento della ricerca biomedica del SSN (project PNRR-MAD-2022-12376546 – DIABETHIC study - "Cardio-metabolic diseases in immigrants and ethnic minorities: from epidemiology to new prevention strategies" – CUP D15E22000760003).

**Competing interests:** The authors have declared that no competing interests exist.

three pillars of type 2 diabetes control (diet, physical activity, and therapeutic adherence). To co-create the intervention, interviews, focus groups and role-plays were conducted with patients and immigrants, healthcare workers and representatives of social services. Identified barriers were ranked according to priority and actionability. Given different health promotion practice in the four centres, the intervention was standardised by function (dietary counselling, culturally tailored information materials, access to cultural mediation, training in effective and reflective communication, individual and group meetings) rather than by form (operators involved, protocols and timeframes), which was defined locally by feasibility and by contrasting usual health promotion. (Trial registration: ClinicalTrials.gov ID NCT06131411).

## 1. Introduction

### 1.1. Background and rationale

**1.1.1. Health problem.** Diabetes is one of the most common chronic diseases in high-income countries. In Italy, its prevalence is more than 6% in the general population and is over 20% in those aged 50 or more. The burden of disease attributable to high blood glucose in Italy is 665.000 DALY (557.000 to 792.000) in 2021 (DALY are defined as the sum of years of life lost and years lived with disability) [1]. A large part of the health system resources is dedicated to diabetes control and the care of diabetes complications.

The burden of diabetes disproportionately affects immigrants, in Italy as well as in most Western European countries [2]. When considering the different age distribution, diabetes prevalence is much higher in immigrants than in native populations, particularly in communities coming from South Asia, Africa, and, with less strength, the Caribbean [3–5]. Furthermore, different studies showed poorer glycemic control [6] and higher incidence of diabetes complications, and cardiovascular and kidney diseases.

There is a complex network of causes behind the higher prevalence and the worst diabetes control in immigrants. Despite there are pieces of evidence that the high prevalence in some ethnic groups may be due to genetic factors, the variability across hosting countries and care systems, as well changes in disease control with changes of diet and lifestyles after the arrival in host countries proof that most causal factors are modifiable.

Potential factors underlying the high risk of type 2 diabetes and its related complications in migrants are multifaceted, including pre and post migration factors. Pre-migration factors include intrauterine growth, parental socioeconomic status [SES], health behaviors, while post-migration factors are the contextual factors in the host countries, lifestyle changes, health systems and related policies. It has been hypothesized that all of these factors can influence socioeconomic circumstances, behavior and biological factors, access to healthcare, physical and psychosocial stress and epigenetics upon migration, which in turn affect insulin secretion and action and subsequently type 2 diabetes risk [7]. Finally, also the hard experience often experimented during journeys can play a role, both directly, generating post-traumatic stress disorders (PTSD) and indirectly, determining the adoption of risky behaviors such as smoking, alcohol intake, or sedentary lifestyles as coping strategies.

Several studies investigated the barriers to effective care and prevention, including providing pharmacological therapy to control glycaemia and lifestyle counselling. Barriers include cultural and linguistic barriers making it difficult to navigate the health system and access chronic care services, inability of the health service to provide tailored health promotion, and logistical and economic difficulties limiting the time for care, physical activity, and purchasing

appropriate foods. These barriers exist even in universal health services when facing newly arrived immigrants. Furthermore, undocumented immigrants have no access to chronic disease care in most countries [8,9].

### 1.1.2. Available interventions and their limits.

In exploring the literature on elements of complex interventions we tried to take into account their sociocultural acceptability and feasibility through the questions suggested by Booth A. et al 2019 in their analysis of an alternative framework for qualitative evidence syntheses for exploring the effects of complex interventions [10].

Recent systematic reviews showed that few interventions are effective in removing these barriers and in improving diabetes control in immigrants [11,12]. Tested interventions usually target diet, physical activity, and compliance with therapeutic plans. Culturally appropriate diabetes health education in ethnic minority groups showed significant improvements in diabetes control. Complex interventions, acting on more factors at both individual and environmental levels, showed a higher probability of success [11,12]. Group-based intervention in people with type 2 diabetes results in improvements in clinical and non-clinical outcomes [13] and it also seems a promising approach for people with a migration background [14].

The literature highlights some recurrent limits in the proposed interventions. Some interventions often are not intensive enough to be effective in changing the environment (the family or the community) to make it easier to adhere to the diet, physical activity and therapeutic recommendations. On the opposite, other interventions are too intensive and invasive to be acceptable and scalable. It can be challenging to tailor the proposed actions on the patient's individual and cultural needs, time and economic constraints but the effectiveness of interventions that take precisely these aspects into account is noticeable [15,16] and co-creation strategies (including patients' involvement) can help to achieve this goal [15].

### 1.1.3. Methodological issues.

The scarcity of evidence-based effective interventions is also due to the difficulties in conducting methodologically sound studies. Testing the efficacy of single components of a complex intervention, with a reductionist approach in which the intervention and the control arm differ only for a specific characteristic will probably produce a too small contrast between the two arms to detect an effect. Furthermore, the reductionist approach will prevent measuring the synergic effects emerging from the interactions between different components of a complex intervention [17].

On the opposite side, randomized trials on complex interventions as whole are difficult to design and conduct. Individual randomization has intrinsic limits in measuring the components of the intervention targeting the environment. Furthermore, a full commitment of the professionals involved is crucial for the effectiveness of culturally sensitive interventions but is difficult to obtain when the intervention is randomly applied only to one-half of their patients. Cluster randomized trials can partially overcome these issues; nevertheless, they are difficult to conduct under the current legislation regulating clinical research and data treatment in Italy. Finally, standardizing the intervention, as usually required in clinical research, would be unfeasible in multicenter studies, because complex interventions need to be adapted to the context [18]. Thus, a standardized complex intervention would probably be less effective, feasible and scalable than a context-adapted ones, reducing the external validity of the study [19–21].

These considerations led us to evaluate different kinds of interventions according to their validity, the attendance to co-creation principles, and the possibility of evaluating them (Table 1).

The first scenario plans to apply the same intervention to all centers. This would lead to difficulties in defining a contrast with usual care in centers where the intervention would be very similar to what is already in place. The transferability of the results (external validity) would not be analyzable as each center had the same intervention. The intervention would not adapt

**Table 1. Possible scenario of intervention and implication for analyses.**

| Scenario | Internal validity (Contrast) | External validity (results transferability) | Attendance of co-creation principles | What is possible to evaluate? |
|---|---|---|---|---|
| 1: Same intervention in each center | No possible analyses (there will be centers in which the intervention is the usual care). | No | No | Nothing |
| 2: Incremental intervention (same increase in each center) | Analyses in terms of attributable risks, heterogeneity difficult to handle. | Reduced. Differences among centers due to different baseline. | Partial | Same specific additional elements compare to usual care. |
| 3: Incremental intervention (increase proportional to the standard of care in each center) | Analyses in terms of attributable and relative risks. | Yes | Yes, increased internal and external validity of the process. | Additional elements. Methodology to develop an intervention through a co-creation process. |

to the context on the basis of a co-created process but it would be implemented by default. It would be difficult to evaluate the effectiveness of the intervention in the absence of contrast within centers and between centers. These limits would be partly overcome with an uniform among centers and incremental intervention, but they are completely overcome only with an incremental intervention with an increase proportional to the standard of care in each center.

An incremental intervention, whose components increase proportionally to the standard of care in each center seems to us the better solution.

**1.1.4. Co-creation.** Co-creation, in the context of a public service design, refers to a process in which the provider collects input from final users that plays a central role in defining the service and how to deliver it [22]. European commission recommends adopting co-creation in designing services to the citizens to bridge the gap between their needs and the public services [23]. A Co-created approach has been applied in health services and particularly to design interventions for chronic care where the goal is improving patient quality of life in the long period through self-medication and adherence to treatments [24]. It has been also proposed to offer services to the hard-to-reach or hard to follow up populations [25]. The co-creative process, through the involvement of final users and providers, should facilitate the design of tailored, cultural sensitive, feasible, and acceptable interventions. Nevertheless, balancing the need to adopt only evidence-based interventions and the patients' requests requires a clear distinction between what is the object of co-creation and what is a matter of scientific debate, and this is not an easy task [26–28].

## 1.2. Objectives

The aim of the study is to evaluate the efficacy of a co-created, culture-sensitive intervention to promote a healthy diet, and physical activity, and to improve compliance to therapeutic protocols in immigrants with type 2 diabetes.

## 1.3. Trial design

We will conduct a multicenter randomized controlled trial comparing the effectiveness of a tailored co-created intervention in immigrants with type two uncontrolled diabetes in the context of the project "Cardio-metabolic diseases in immigrants and ethnic minorities: from epidemiology to new prevention strategies" funded by the European Union (NextGenerationEU). Four centers take part in the project: Careggi University Hospital (Florence), Local Health Authority of Reggio Emilia, National Institute for Heath, Migration and Poverty (INMP, Rome), Dulbecco University Hospital (Catanzaro). The main outcome is glucose control measured as glycated hemoglobin (HbA1c). We will evaluate a complex intervention

based on three pillars, diet, physical activity, and adherence to therapeutic protocols, vs. the usual care (S1 Table). The components of the complex intervention will be standardized by defining their functions [18,19], as opposite of by their form, giving the opportunity to adapt the form of the actions and tools to the context in each center. Usual care is also center-dependent. Therefore, we will also adapt the intervention according to the effort needed to increment the usual care to the intervention, i.e., each center will include in the intervention arm the components that will be affordable with a similar incremental effort (Table 1).

## 2. Trial procedures and protocol

### 2.1. Participants, interventions, and outcomes

**2.1.1. Study setting and study schedule.** The study will be conducted in the primary care clinics specialized for the care of diabetes of the centres taking part of the project. The figure one showed the study schedule (Fig 1).

**2.1.2. Eligibility criteria.** Adult (>=18) individuals with diabetes with an immigration background with a new diagnosis of diabetes or uncontrolled diabetes (HbA1c%>=8). Exclusion criteria include patients who will not provide the informed consent, patients aged under 18, patients with HbA1c ≤ 8% in the last assessment within 24 months before the visit, patients with severe psychiatric disorders, pregnant women, critical illness, impaired cognitive or physical ability that could make the intervention not feasible, as judged by clinical staff members.

**2.1.3. Co-creation methods and description of the intervention.** The intervention has been defined through a two-level co-creation process, a centralized level (coordinated by the Reggio Emilia Centre) that established the general principles and the possible components of a complex intervention, and a local level that decided which components were more useful and feasible in the local context and how they should be implemented. The result was a complex multicomponent intervention, in which each component was standardized by its function, tailored and culturally sensitive, context adapted and in which the contrast vs. the usual car was defined by similar incremental effort.

The co-creation process included different strategies to involve and collect the voices and angles of users and providers. We adopted a co-creation approach (involving stakeholders and patients to identify the barriers and solutions). The strategies adopted in the centralised phase of co-creation included:

- Focus group, conducted in Reggio Emilia, with providers and lay persons from immigrant communities. Providers involved were selected among workers of municipality and cultural mediation services, third sector organizations caring for undocumented migrants, primary care services caring for people with diabetes, and public health services involved in health promotion and primary prevention. Lay people were contacted through participating providers and selected based on formal or informal knowledge of the diet and lifestyle of the communities involved. The groups worked on the analyses of existing barriers and facilitators in access to care and prevention and in adherence to behavioural and therapeutic recommendations.

- Interviews of the patients and healthcare workers in diabetes clinics. In each centre, patients in the waiting rooms of diabetes clinics were asked to take part in an interview involving five open-ended questions on possible barriers and facilitators of diabetes control both related to their daily lives and to the services of the healthcare system, developed on an ad hoc basis by the qualitative research team. The health care workers were invited to the same interview, suitably modified.

- Laboratory to match proposed solutions, literature evidence, and feasibility/sustainability. Informed by the two systematic reviews that we used for guidance.

| | STUDY PERIOD | | | | | |
|---|---|---|---|---|---|---|
| | Enrolment | Allocation | Post-allocation | | | Close-out |
| **TIMEPOINT** | *0* | *0* | *1 m\** | *6 m\** | *12 m\** | *12 months* |
| **ENROLMENT:** | | | | | | |
| **Eligibility screen** | X | | | | | |
| **Informed consent** | X | | | | | |
| **Allocation** | | X | | | | |
| **INTERVENTIONS:** | | | | | | |
| *Co-created intervention* | | | ●———● | | | |
| *Usual care* | | | ●———● | | | |
| **ASSESSMENTS°:** | | | | | | |
| *CORE QUESTIONNAIRE* | X | | ●———————● | | | X |
| *IPAQ-SF* | X | | | | | X |
| *MedDietScore* | X | | | | | X |
| *DMTAS* | X | | | | | X |
| *Anthropometric measures* | X | | | | | X |
| *Blood & urine Biomarkers* | X | | | | | X |

*m=months.

°The complete assessment will be performed at baseline and 12 months follow-up visits. Information on the health promotion activities attended by each patient in both groups will be also recorded. Finally, routinely collected data on blood and urine biomarkers, anthropometric measures and changes in diabetes related lifestyles and therapeutic regimen during the study period will be also recorded and included in the analysis.

**Fig 1. Study schedule of enrollment, interventions, and assessments according to SPIRIT 2013 requirements.**

The results of this step are summarized in the Fig 2, where we adopted a simplified DPSEEA model to conceptualize how we addressed actionable barriers or facilitators with components of the complex intervention.

The local phase of the co-creation was based on local laboratories with key persons and providers that analyzed the current usual care, the available resources, and the components of

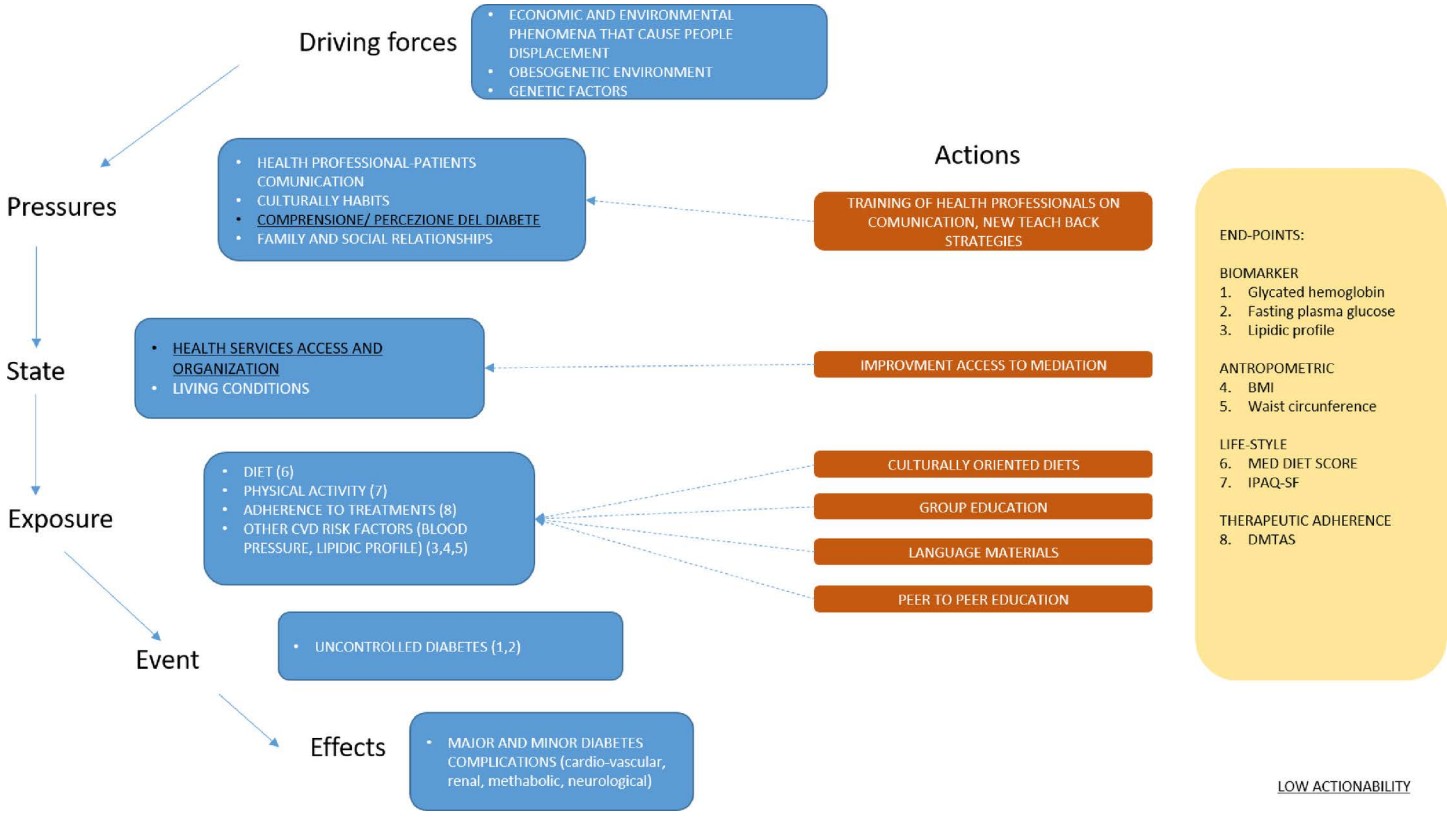

**Fig 2. Conceptual framework of the intervention.**

the complex interventions defined at the centralized level for their feasibility at local level. The results of this phase are summarized in the (S1 Fig).

Finally, the results of the local laboratories were discussed in a plenary session of the trial steering committee in Rome (October 13th, 2023) to compose the best complex intervention in each local context given the available additional resources provided by the experimental project and the existing services. Results of the co-creation process are summarized in the Table 2.

**2.1.4 Usual care.** As mentioned above, the usual care was different in each centre. An analysis of the process was conducted in each centre through interviews and laboratories. During this analysis, it became clear that the implementation of the interventions would necessarily change some characteristics of the usual care; the local researchers made the effort to make explicit any unintended and unavoidable change of the usual care. The characteristics of the usual care are reported in Table 2.

**2.1.5 Outcomes.** The primary outcome is the change of HbA1c 12 months after recruitment.

The secondary outcomes are the changes in:

- Anthropometric measures (BMI, waist circumference)

- Dietary habits

- Physical activity habits

- Lipid profile

- Compliance with individual therapeutic protocols

**Table 2.** Intervention's components by center and topic.

| | Florence Usual care | Florence Intervention | INMP Usual care | INMP Intervention | Reggio Emilia Usual care | Reggio Emilia Intervention | Catanzaro Usual care | Catanzaro Intervention |
|---|---|---|---|---|---|---|---|---|
| **Medical examination** | 30-minutes first diabetological examination. 20-minutes diabetological follow-up examination. | As usual care. | Oupatient internist medical examination at NIHMP (standard duration: 30 minutes). Screening test for diabetes complications provided in the same facility (eye examination with ocular fundus, vessels and heart internist ultrasonography and ecocolor doppler, blood tests, electrocardiogram). | Oupatient internist medical examination at NIHMP (duration: 40 minutes). Screening test for diabetes complications provided in the same facility with cultural mediator and monitoring of compliance during follow-up visits. | First appointment upon the request issued by the fanily doctor according to the properties shared by the regional document about appropriateness. Diabetological follow-up appointments every 6–8 months or according to the clinical need. | As usual care. | Absence of dedicated clinical pathway for migrants. Poor attendance of migrants at the hospital facility. Regular hospitalization, DH, oupatient clinic. | Creation of clinical dedicated pathway for migrants (PAC, DH). |
| **Mediator** | Permanent Chinese mediator once a week. Activation of other languages mediators on demand for diabetological examinations. Mediator not provided for individual nutrition-focused physical exam with dietitian. | Cultural mediator provided for individual nutrition-focused physical exam with dietitian, too. | Cultural mediator provided for the following languages: Bengali, Somali, Arabic, French, Albanian, Romanian, English, Spanish, Pidgin English. | Cultural mediator provided for individual nutrition-focused physical exam with dietitian, too. | «Random» activation of the mediation through mail on demand. | Ensure the continuity of the mediation service (writing the name of the mediator on medical record and ask for the same person next time; if necessary, involvement of mediators during group sessions. | Absence of cultural mediators (possibility of subcontract with cooperative). | Activation of cultural mediators. |
| **Material** | Glycemic diary and language informative materials. | Culturally-oriented language informative materials. | Language glycemic diary and language informative materials (English, Spanish). | Culturally-oriented language informative materials. | | Use of culturally-oriented language materials. | | |

*(Continued)*

**Table 2.** (Continued)

| | Florence | | INMP | | Reggio Emilia | | Catanzaro | |
|---|---|---|---|---|---|---|---|---|
| | Usual care | Intervention | Usual care | Intervention | Usual care | Intervention | Usual care | Intervention |
| **Diet** | One appointment with a dietitian after medical examination (standard duration of 40 minutes) + follow-up visits (standard duration of 15 minutes). Poor patients attendance due to lack of a cultural mediator. | Individual intervention: individual nutrition-focused physical exam with permanent cultural mediator (duration: 30 minutes) + 3 follow-up visits (duration: 30 minutes). Group education: at least one group session focused on cooking and nutritional education with cultural mediator. 6–8 patients involved in each session. Tools and materials: food portions quantification tools (bowls, glasses), structured dietary history, language informative materials. | General nutritional recommendations. | Individual intervention: individual nutrition-focused physical exam with a nutrition specialist doctor and permanent cultural mediator. Group education: group sessions with a nutrition specialist doctor and cultural mediator focused on eating habits, physical activity, health education aimed at preventing complications. | One appointment with a dietitian after the first medical examination or on demand (acute glycemic decompensation) (standard duration of 20 minutes). Monthly proposed group therapeutic education sessions, but most of the patients with migration background don't partecipate to it. | Individual intervention: individual nutrition-focused physical exam after the first medical examination (or in occasion of the recruitment/decompensation) at T0 and at least a second NFPE within 6 months from the recruitment (T6) (first visit duration: 40 minutes, 20 minutes as in usual care + 20 minutes of intervention; following visits duration: 20 minutes). Group education: identification of common needs during individual interviews and organize group sessions focused on cooking and nutritional education (at least 1 group session). Tools and materials: intercultural calendar, food portions quantification kit, structured dietary history, language informative materials. Advance of the nNFPE contents supported by informative flyer. | Clinical nutrition department rarely involved in the management of diabetic patients. | Individual nutrition-focused physical exam for all the diabetic patients. |
| **Physical activity** | General recommendations. | Individual intervention: customized lifestyle promotion recommendations (including physical activity) during individual nutrition-focused physical exam with a dietitian. Group education: at least one group session focused con lifestyle and physical activity with cultural mediator. 6–8 patients involved in each session. | General recommendations. | Group education: group sessions with a nutrition specialist doctor and cultural mediator focused on eating habits, physical activity, health education aimed at preventing complications. | Generic recommendations during medical examination or first appointment with a dietitian. | Individual intervention: customized lifestyle promotion recommendations (including physical activity) during individual nutrition-focused physical exam with a dietitian. Group education: group sessions focused on lifestyle and physical activity, smoking cessation, in line with the needs identified during individual interviews (at least 1 group sessione. Tools and materials: tailored conversation map. | | Dedicated operators involved. |

*(Continued)*

**Table 2.** (Continued)

| | Florence | | INMP | | Reggio Emilia | | Catanzaro | |
|---|---|---|---|---|---|---|---|---|
| | Usual care | Intervention | Usual care | Intervention | Usual care | Intervention | Usual care | Intervention |
| **Education** | | Operators training organized by other centers. | Cultural mediators training. | Involved cultural mediators specific training. | | Teach back and effective communication; mediators training; tools sharing. | | Creation of a multidisciplinary team. |
| **Therapeutic adherence** | | Use of techniques learned during the training. | Therapeutic recognition during medical examination. | Use of techniques learned during the training. | Investigation during medical examination about therapeutical adherence. | Use of «teach back» technique and effective communication skills; focus on connection between lifestyle and pharmacological treatment during both individual and group sessions. Report on electronic patient record. | | Cultural mediators involved to motivate patients. |
| **Health education** | Group therapeutic education sessions, but most of the patients with migration background don't partecipate to it. | Group education: at least one group session focused on lifestyle/physical activity and therapeutic education with cultural mediator: 6–8 patients involved in each session. | | Group education: group sessions with an internist and cultural mediator focused on eating habits, physical activity, health education aimed at preventing complications. | Nursing training course at the request of diabetologists. Monthly proposed group therapeutic education, but most of the patients with migration background don't partecipate to it. | | | Specific meetings with patients. |

The outcomes and the covariate measurements will be at recruitment and at follow up (Fig 3). Between the baseline and the 12 months evaluation clinical data noted during routine diabetology visits are recorded, regardless of the randomization arm.

**2.1.6. Sample size.** To explore the efficacy of a culturally adapted diabetes education model in improving health literacy and self-care (primary endpoint) in immigrant patients with type 2 diabetes, the intervention is expected to provide a mean 0.5% higher reduction in the concentration of HbA1c in the intervention group compared to the usual care at a 12-month follow-up, with a positive effect on the risk of CVD events [29,30]. Through a change in health-related behaviours, it is also expected to reduce overweight and increase vegetable intake and physical activity, with a positive impact on future risk of non-communicable diseases and quality of life [1]. To have a power of 80%, considering an alpha of 0.05 [31], in order to detect a minimum significant reduction of 0.5% in the glycated haemoglobin in the intervention group compared to the control group at least 200 participants are needed (considering a SD in both group of 1.3) [32].

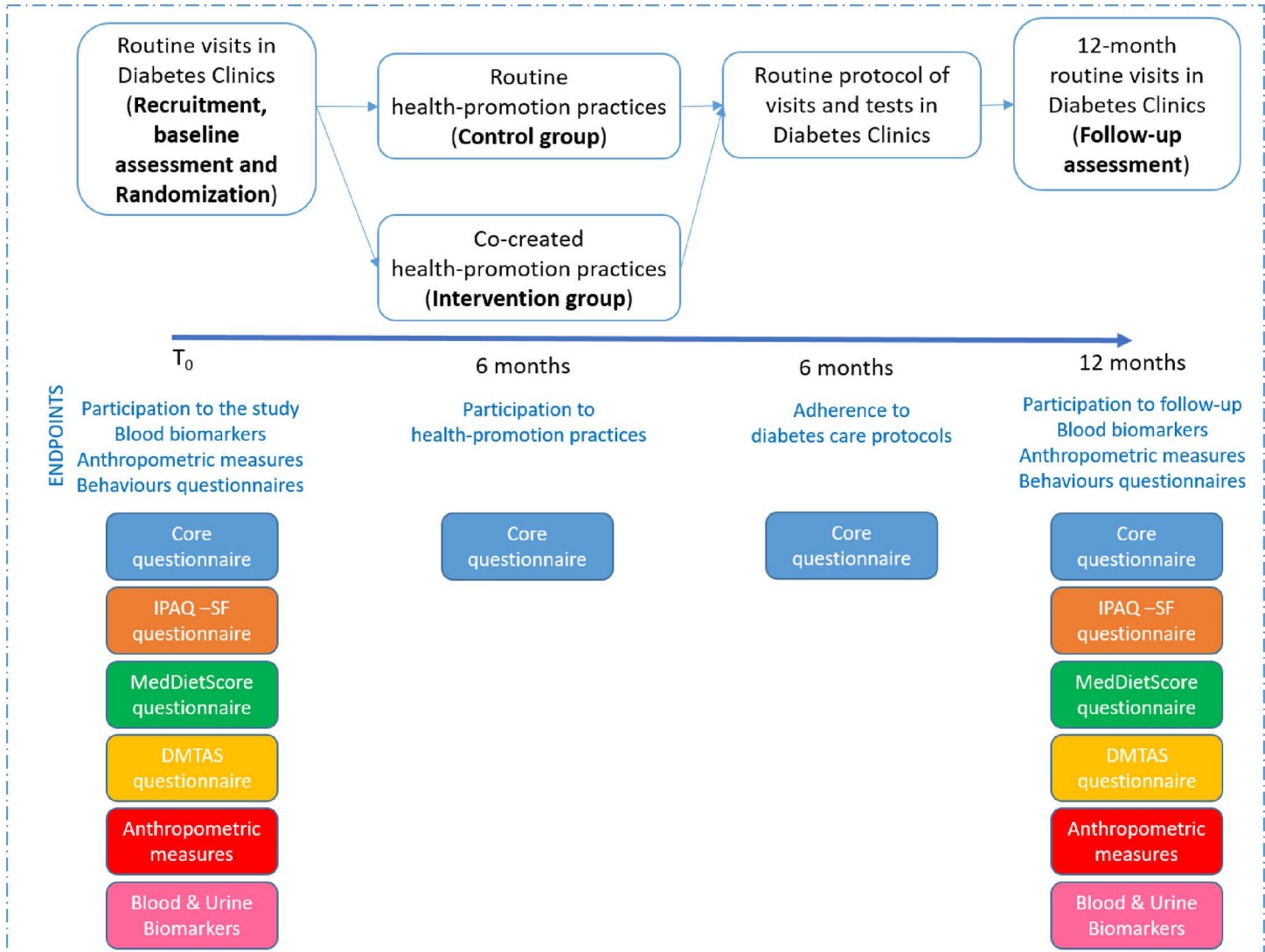

**Fig 3. Study timeline.**

**2.1.7. Recruitment.** Participants will be asked to participate when attending a visit at one of the participating clinics. Whether low participation rates will occur, a preliminary eligibility assessment on medical records of Diabetes clinics will be performed, and potentially eligible patient will be actively contacted. Pre-recruitment eligibility criteria assessment from third sector organizations caring for undocumented migrants will also support active recruitment. Consecutive enrolment of potentially eligible patients started on the 7th of November, 2023 and will last until reaching the anticipated sample size.

**2.1.8. Assignment of interventions.** Block randomization (1:1 ratio) will be computerised and conducted in each centre to balance the ethnic composition of the intervention and control groups. The random sequence was generated using the REDCap™ Random Sequence Generator. Members of the same household, whether identified as cohabitant during the recruitment visit, will be assigned at the same arm. The person who enrolls will be blinded to the random sequence until each patient will sign the informed consent. The randomization arm will not be masked to the participant nor to the investigator.

We planned two strategies to reduce the potential performance bias arising from the lack of blinding of participants and investigators. The first was the choice of an objective outcome as primary outcome of efficacy in our study, namely the change of HbA1c 12 months after recruitment. The second was the involvement of experienced health professionals for the performance of the intervention, that will be sensitize and trained on good practices for clinical trials.

Nevertheless, the qualitative information that will be available during the co-evaluation phase of the project will support us to assess the potential implication of performance bias on the trial's results.

**2.1.9. Data collection.** Biomarkers will be assessed through analysis of blood and urine samples collected following standard procedures of the Diabetes clinics and analysed for glycaemia, by the authorised clinical laboratories of each recruiting centre.

Questionnaires used to assess dietary habits and physical activity will be the Mediterranean Diet Score, (MedDietScore) [33] and the International Physical Activity Questionnaire - Short Form, IPAQ-SF [34].

Anthropometric measures will be collected following standard procedures during routine visits in Diabetes clinics with validated weight and height scales and bioelectrical impedance analysis (BIA).

Therapeutic adherence will be assessed at baseline and follow up visits using the Diabetes Mellitus Treatment Adherence Scale (DMTAS) [35].

Data will be recorded through the REDCap™ web application provided by the Principal Investigator to all the recruiting centres. Data access and management will be performed in compliance with the Italian and European privacy regulations.

**2.1.10. Statistical plan.** Descriptive statistics will be calculated for baseline characteristics. The analysis will be intention-to-treat. Before/after variations of the glycated haemoglobin, lipid profile and renal function biomarkers, and anthropometric measure will be computed and standardized if opportune. Paired and unpaired tests will be used to assess the effects of the intervention and to analyse before/after within-group and between-group differences and changes in glycated haemoglobin. Changes in lifestyle habits (both related to diet and physical activities) will be described in terms of positive or negative changes.

Multivariate linear regression models will be used to analyse the variation of anthropometric measures and assessed biomarkers.

Multilevel linear models will be performed taking into account the influence of the centre on intervention efficacy.

We will perform a mediation analysis to understand which part of the changes in outcomes (glycated haemoglobin, lipid profile, and anthropometric measures) is attributable to changes

in physical activity, changes in diet, adherence to therapies, and other direct or unmeasured effects of the intervention.

The statistician will not be blinded to group assignments.

The statistical significance level will be set at 5%, and all analyses will be performed by using Stata 16 or SPSS 28.

**2.1.11.  Subgroup analysis.**  Subgroup analyses will be performed according to weight status, gender, ethnicity, and socioeconomic level.

## 2.2.  Patient and public involvement

Patients and the local communities were actively involved in the development and design of the intervention as described in the "Co-creation methods and description of the intervention" section. Patients, immigrants, healthcare professional and representatives of social services will continue to be involved in the co-evaluation and dissemination phase. They will support the interpretation of study results, drafting plain language summaries, and co-presenting findings to lay people at community level.

## 2.3.  Ethics and dissemination

Diabethic trial is part of the "Cardio-metabolic diseases in immigrants and ethnic minorities: from epidemiology to new prevention strategies" project has been approved by the Italian Ministry of Health for funding to the call Piano Nazionale di Ripresa e Resilienza (PNRR): M6/C2_CALL 2022, Ministero della Salute, funded by the European Union. The role of the funding source is reported in the call of the Ministry of Health (Bando Piano Nazionale di Ripresa e Resilienza) available at https://www.agenziacoesione.gov.it/comunicazione/piano-nazionale-di-ripresa-e-resilienza/ https://ricerca.cbim.it/Documentazione. The Tuscany Regional Ethics committee "Comitato Etico Regionale per la Sperimentazione Clinica della Toscana - sezione AREA VASTA CENTRO" approved the trial protocol on November 29, 2022. (S1 File)

This trial protocol has been registered with the ClinicalTrials.gov Registry (ClinicalTrials.gov ID NCT06131411). The protocol was framed according to the SPIRIT Reporting guidelines (S1 Checklist) [36].

The trial results will be available in 2025 and will undergone to a co-evaluation process involving all the stakeholders that contributed to the co-creation process. The co-evaluation of trial results aims to include all the different perspective to assess the actual impact of the whole process on patients and healthcare services. The co-evaluation may also support the implementation phase of the intervention, highlighting putative determinants of effectiveness and effect-modification that should be taken into account when adapting the intervention in different contexts.

The results will be published in a peer-reviewed medical journal.

**2.3.1.  Informed consent.**  The eligible patients will be informed on the study and those interested in participating will be asked to sign informed consent during routine visits at Diabetes clinics. Patients aged < 18 years or unable to provide a personal informed consent are not eligible for the study.

## 3.  Discussion

In designing this protocol, we tried to address both the issues of defining an acceptable and scalable intervention and to construct a study that could measure its efficacy. We acknowledged that a co-created and context-adapted complex intervention could not be standardized in its form across centres and also standardizing it by all the functions of the intervention

components would be unrealistic. Furthermore, we observed an extreme heterogeneity in usual care across centres. Therefore, one of the main problems was how to balance the two needs: being context-adapted and measuring the efficacy of a reproducible intervention. The result was a careful conceptualization of what was the object of the evaluation to build an experimental design that could compare the novelty of the intervention in contrast to the existing usual care (Table 1). Indeed, during the evaluation phase, we would take into account the heterogeneity at function and incremental effort levels, which is expected to be lower than the heterogeneity that will be observed at form level.

### 3.1. Co-creating the intervention

The main limits that we identified in previously proposed interventions were the lack of tailoring, scarce cultural-sensitivity, too-intensive interventions that would be not acceptable to the patients and not sustainable by the providers or interventions that were not comprehensive enough to address both the environment and the individual [37,38].

We tried to overcome some of these limits by adopting a co-creative process to design and evaluate the intervention.

Several previous studies on interventions for the management of chronic conditions included co-creation approaches with dissimilarities regarding stakeholders, phases of the research process involved, and methods. Regarding the selection of stakeholders to be involved, previous studies included in the co-creation phase the same patients recruited for testing the interventions [24,39] We decided to involve different populations to assess the external validity of the results of the co-creation phase, with a transcultural approach suggested by the anthropologists of our qualitative research team.

Regarding the phases of the research process, previous studies usually applied a co-creation approach in one phase only, mainly for the definition of the intervention or its evaluation, in terms of acceptability or feasibility [24,27,28,40]. We decided to plan both co-development and co-evaluation phases with the aim of developing an effective intervention and to provide a comprehensive assessment including the stakeholders' perspectives and increasing the researchers accountability.

Regarding the methods, the main differences between previous studies were related to the background of professionals involved in the co-creation phase and in the modes of stakeholders' engagement (i.e., face-to-face, on-line, or blended) [28,39]. In our study, two anthropologists and one sociologist with background in qualitative research coordinated the co-creation phase, and provided support to keep a transcultural approach across all the research process phases. Our preferred mode of involvement of stakeholders was face-to-face, given potential limitations on accessibility to on-line tools of the target population. Indeed, a blended mode may be a promising alternative approach.

In our study, the co-creative approach became the main characteristic of the intervention and the object of the evaluation.

During the process, we experienced many of the difficulties that caused the limits of previous interventions: only minimal changes in the organization of the service providing were considered sustainable; intensive counselling was too time-consuming and lifestyle changes were considered unacceptable by patients; including undocumented immigrants was considered difficult, if not unfeasible, for both administrative and logistic issues; in particular, although the Italian national health service is universalistic and guarantee access to everybody, including undocumented immigrants, it is difficult to imagine to take care these subgroups of population, often only in transit in a specific city or in any case, not confident to receive health care in a same healthcare facility due to their illegal status

All these constrains reduced the choice of possible components to be included in the intervention in most centres. The co-creation process made it easy to identify these barriers and probably magnified the problems. Nevertheless, the process allowed us to find shared solutions.

### 3.2. Creating the contrast between the intervention and usual care

The analysis of the usual care in each centre showed an extreme heterogeneity. This heterogeneity represents the differences existing in diabetes care across different public clinics in Italy, despite the health system being substantially similar in its funding and organization. Indeed, we noted that the differences across the country in providing health promotion and in proactive care of chronic disease are larger than those observed in acute care. Several factors may contribute to this heterogeneity. In health promotion guidelines often suggest (conditional recommendations) a set of possible interventions, not mutually exclusive; furthermore, in proactive care of chronic conditions the way of providing services is often as important as the service itself.

Having different usual care protocols in the control arm is a problem when conducting a trial. We tried to conceptualize the different options in terms of how to standardize the intervention and the usual care across centres and how this would affect what we are measuring in terms of causal inference, the internal and external validity of the study, and on the feasibility of the study. The last point is not only an accidental constraint that we have to overcome to make a good trial, it may also be a proxy of the scalability and feasibility of the proposed intervention in each context. Table 2 summarises this point. In fact, in the presence of such variability of the usual care, the feasibility of a given multicomponent intervention would be different in each centre: some components of the intervention are yet current practice in one centre, while in another can be considered almost unfeasible. Furthermore, what would be a feasible intervention in all centre, would be less than what already offered as usual care in the centres with a more structured usual care. The solution we found was to standardize the intervention on the incremental effort, i.e., in each centre, the intervention should require similar additional resources. Unfortunately, each component did not require the same additional amount of resources in each centre. In fact, in most cases, in centres where the usual care includes only basic services, implementing some components required much larger efforts than in centres already able to provide more complex services.

### 3.3. Lesson learnt/Conclusions

Designing a multicentre trial to evaluate a complex intervention defined through a co-creative approach introduces further complexity and requires a new conceptual framework. We tried to build such a framework to understand how to deal with existing differences and with the intrinsic tailoring of a co-created intervention. We started adopting a standardization of the components of the intervention by function instead of by form, as suggested by Hawe et al [19]. Focussing on feasibility and scalability [18], we also introduced the concept of standardizing the intervention also by the incremental resources needed.

The conduction of the trial and its results will show if our attempts were fruitful.

## Supporting information

**S1 Table. Elements of the intervention by topic and flexibility (degree of freedom of co-creation approach) in the definition.**
(PDF)

**S1 Fig. Results of barriers' analysis by center.**
(TIF)

**S1 File. Study protocol approved by the ethics committee v 1.1.**
(PDF)

**S1 Checklist. SPIRIT checklist.**
(DOCX)

## Acknowledgments

We sincerely thank the patients, citizens, healthcare professionals, and representatives of social services who actively participated in this co-creation process. Their invaluable insights and feedback have significantly shaped the study's design.

DIABETHIC Working group: Paolo Giorgi Rossi, Francesco Venturelli, Laura Bonvicini, Francesca Bononi, Massimo Vicentini, Giulietta Luul Balestra, Giusy Iorio, Luca Ghirotto, Elisa Manicardi, Eles Notari, Simona Bodecchi, Bruna Milli, Silvia Pilla, Giulia Bellei, Miriam Parisi, Alessandro Lo Ioco, Prisco Sbordone, Claudio Bongiorno, Massimo Michelini, Antonella Paola Sacco, Roberta Lunghi, Francesca Palmieri, Alessio Petrelli, Silvia Pierconti, Giovanna Laurendi, Anteo Di Napoli, Martina Ventura, Paola Coletta, Giulia Barbarossa, Alessio Pellegrino, Maria Boddi, Pietro Amedeo Modesti, Irene Vacirca, Cecilia Baccari, Maria Calabrese, Maria Perticone

Working group lead author: **Pietro Amedeo Modesti** (pa.modesti@unifi.it), Alessio Pellegrino, Maria Boddi, Irene Vacirca, Cecilia Baccari, Maria Calabrese (Department of Medical and Surgical Sciences, University Magna Graecia of Catanzaro, Catanzaro, Italy, Medicina dello Sport e dell'Esercizio Fisico, Azienda Ospedaliero-Universitaria Careggi, Florence, Italy), Paolo Giorgi Rossi, Francesco Venturelli, Laura Bonvicini, Francesca Bononi, Massimo Vicentini (Epidemiology Unit, Azienda USL-IRCCS di Reggio Emilia, Reggio Emilia, Italy), Giulietta Luul Balestra, Giusy Iorio, Luca Ghirotto (Qualitative Research Unit, Azienda USL-IRCCS di Reggio Emilia, Reggio Emilia, Italy), Elisa Manicardi, Eles Notari, Simona Bodecchi, Bruna Milli, Silvia Pilla, Giulia Bellei, Miriam Parisi, Alessandro Lo Ioco, Prisco Sbordone, Claudio Bongiorno, Massimo Michelini, Antonella Paola Sacco, Roberta Lunghi, Francesca Palmieri (Diabetology Unit, Azienda USL-IRCCS di Reggio Emilia, Reggio Emilia, Italy), Alessio Petrelli, Silvia Pierconti, Giovanna Laurendi, Anteo Di Napoli, Martina Ventura, Paola Coletta, Giulia Barbarossa (Epidemiology Unit, National Institute for Health, Migration and Poverty (INMP), Rome, Italy), Maria Perticone (Department of Medical and Surgical Sciences, University Magna Graecia of Catanzaro, Catanzaro, Italy)

## Author contributions

**Conceptualization:** Laura Bonvicini, Francesco Venturelli, Alessio Petrelli, Maria Perticone, Alessio Pellegrino, Pietro Amedeo Modesti, Paolo Giorgi Rossi.

**Data curation:** Francesca Bononi, Silvia Pierconti, Giovanna Laurendi, Alessio Pellegrino, Maria Boddi.

**Formal analysis:** Laura Bonvicini, Paolo Giorgi Rossi.

**Funding acquisition:** Francesco Venturelli, Alessio Pellegrino, Pietro Amedeo Modesti, Paolo Giorgi Rossi.

**Investigation:** Francesco Venturelli, Francesca Bononi, Giulietta Luul Balestra, Giusy Iorio, Luca Ghirotto, Alessio Petrelli, Silvia Pierconti, Giovanna Laurendi, Maria Perticone, Alessio Pellegrino, Maria Boddi, Pietro Amedeo Modesti.

**Methodology:** Laura Bonvicini, Francesco Venturelli, Giulietta Luul Balestra, Giusy Iorio, Luca Ghirotto, Alessio Petrelli, Silvia Pierconti, Giovanna Laurendi, Maria Perticone, Alessio Pellegrino, Pietro Amedeo Modesti, Paolo Giorgi Rossi.

**Supervision:** Pietro Amedeo Modesti, Paolo Giorgi Rossi.

**Validation:** Paolo Giorgi Rossi.

**Visualization:** Laura Bonvicini, Francesco Venturelli, Francesca Bononi, Silvia Pierconti, Alessio Pellegrino.

**Writing – original draft:** Laura Bonvicini, Francesco Venturelli, Francesca Bononi, Giulietta Luul Balestra, Alessio Petrelli, Alessio Pellegrino, Pietro Amedeo Modesti, Paolo Giorgi Rossi.

**Writing – review & editing:** Laura Bonvicini, Francesco Venturelli, Francesca Bononi, Giulietta Luul Balestra, Giusy Iorio, Luca Ghirotto, Alessio Petrelli, Silvia Pierconti, Giovanna Laurendi, Maria Perticone, Alessio Pellegrino, Maria Boddi, Pietro Amedeo Modesti, Paolo Giorgi Rossi.

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
