## [Decision Letter · Decision Letter 0]

8 Oct 2024

PONE-D-24-33957A randomized trial to evaluate a complex, co-created, culture-sensitive intervention to promote healthy lifestyles and compliance to therapy in immigrants with type 2 diabetes: a protocol of a multicenter Italian studyPLOS ONE

Dear Dr. Venturelli,

Thank you for submitting your manuscript to PLOS ONE. After careful consideration, we feel that it has merit but does not fully meet PLOS ONE’s publication criteria as it currently stands. Therefore, we invite you to submit a revised version of the manuscript that addresses the points raised during the review process.

Please submit your revised manuscript by Nov 22 2024 11:59PM, If you will need more time than this to complete your revisions, please reply to this message or contact the journal office at plosone@plos.org . Please include the following items when submitting your revised manuscript:

We look forward to receiving your revised manuscript.

Kind regards,

Omid Dadras, MD, PhD

Academic Editor

PLOS ONE

3. One of the noted authors is a group or consortium [DIABETHIC Working group]. In addition to naming the author group, please list the individual authors and affiliations within this group in the acknowledgments section of your manuscript. Please also indicate clearly a lead author for this group along with a contact email address.

4. We note that the original protocol that you have uploaded as a Supporting Information file contains an institutional logo. As this logo is likely copyrighted, we ask that you please remove it from this file and upload an updated version upon resubmission.

Reviewers' comments:

Reviewer's Responses to Questions

**Comments to the Author**

1. Does the manuscript provide a valid rationale for the proposed study, with clearly identified and justified research questions?

Reviewer #1: Yes

Reviewer #2: Yes

2. Is the protocol technically sound and planned in a manner that will lead to a meaningful outcome and allow testing the stated hypotheses?

Reviewer #1: Yes

Reviewer #2: Yes

3. Is the methodology feasible and described in sufficient detail to allow the work to be replicable?

Reviewer #1: Yes

Reviewer #2: No

4. Have the authors described where all data underlying the findings will be made available when the study is complete?

Reviewer #1: Yes

Reviewer #2: Yes

5. Is the manuscript presented in an intelligible fashion and written in standard English?

Reviewer #1: Yes

Reviewer #2: Yes

6. Review Comments to the Author

You may also provide optional suggestions and comments to authors that they might find helpful in planning their study.

Reviewer #1: This study protocol addresses a significant health issue (uncontrolled diabetes in immigrants) using a culturally sensitive and participatory intervention approach for a multi-center clinical trial. The protocol has an explanatory introduction, reviewing relevant literature, and a thorough well-detailed methodology with clear outcome measures. In addition, the protocol is registered and adheres to ethical guidelines, with required checklists provided as supplementary files.

In the discussion part, the authors acknowledge that strict replication and generalization of the study results in other settings and centers may be difficult due to the complexity of the intervention, the need for local adaptations, and differences in usual care.

However, there are areas where the manuscript could be improved:

1- The S3 Appendix Figure (Results of barriers’ analysis by center) is neither explained nor mentioned in the manuscript text and should be referenced and explained for clarity.

2- While the current discussion part deals with the limitations of prior interventions, I recommend the authors explore more in-depth the comparison between this protocol and specific past studies, which would enhance the clarity and strength of the argument. Briefly referencing key studies and clearly highlighting how their co-creation approach differs from previous studies with comparable methods, as well as its potential impact on diabetes management efficacy.

3- The study states that the intervention will not be blinded to participants or investigators, which increases the risk of performance bias, particularly when behavioral interventions are being evaluated. It would be beneficial to talk more about ways to lessen this bias.

Reviewer #2: Authors plan to conduct a multicentre randomized controlled trial (RCT) to assess the effectiveness of a co-created health promotion intervention for immigrants with T2D. They will recruit 200 participants from four centers randomized into two arms: health promotion practice and experimental health promotion intervention. The focus will be on diet, physical activity, and therapeutic adherence.

1. Table 1 is unclear to this reviewer. Some more explanations for table 1 are needed.

2. Data will be collected from four centers and it seems that the intervention will not be identical. If so, how to address the potential heterogeneity?

3. Line 263. Typo: enrolls not enrols

4. Why won’t the statistician be blinded to group assignments? is there any reason to be unblinded?

5. What questionnaire will be used for interview?

7. PLOS authors have the option to publish the peer review history of their article (what does this mean? ). If published, this will include your full peer review and any attached files.

**Do you want your identity to be public for this peer review?** For information about this choice, including consent withdrawal, please see our Privacy Policy .

Reviewer #1: No

Reviewer #2: No

---

## [Author Response · Author response to Decision Letter 1]

19 Nov 2024

RE: ok, done

RE: the article presents a study protocol. There are no data available.

3. One of the noted authors is a group or consortium [DIABETHIC Working group]. In addition to naming the author group, please list the individual authors and affiliations within this group in the acknowledgments section of your manuscript. Please also indicate clearly a lead author for this group along with a contact email address.

RE: Ok, we have done it

4. We note that the original protocol that you have uploaded as a Supporting Information file contains an institutional logo. As this logo is likely copyrighted, we ask that you please remove it from this file and upload an updated version upon resubmission.

RE: we removed the logo from the document

RE: we’ve checked the references

Reviewers' comments:

Reviewer #1: This study protocol addresses a significant health issue (uncontrolled diabetes in immigrants) using a culturally sensitive and participatory intervention approach for a multi-center clinical trial. The protocol has an explanatory introduction, reviewing relevant literature, and a thorough well-detailed methodology with clear outcome measures. In addition, the protocol is registered and adheres to ethical guidelines, with required checklists provided as supplementary files.

In the discussion part, the authors acknowledge that strict replication and generalization of the study results in other settings and centers may be difficult due to the complexity of the intervention, the need for local adaptations, and differences in usual care.

RE: thank you for the comment and the careful revision

However, there are areas where the manuscript could be improved:

1- The S3 Appendix Figure (Results of barriers’ analysis by center) is neither explained nor mentioned in the manuscript text and should be referenced and explained for clarity.

RE: Thank you for the comment. We got confused with the citations of the supplementary materials. Now, we correctly number supplementary files. S2 Appendix Figure is the result of the local analysis reported in lines 221-223 (page 9).

2- While the current discussion part deals with the limitations of prior interventions, I recommend the authors explore more in-depth the comparison between this protocol and specific past studies, which would enhance the clarity and strength of the argument. Briefly referencing key studies and clearly highlighting how their co-creation approach differs from previous studies with comparable methods, as well as its potential impact on diabetes management efficacy.

RE: thank you for the comment. We expanded the discussion on the dissimilarities in the co-creation approach of our study compared to previous studies. The main differences were related to stakeholders, phases of the research process involved, and methods. (see lines 364-383, page 17).

3- The study states that the intervention will not be blinded to participants or investigators, which increases the risk of performance bias, particularly when behavioral interventions are being evaluated. It would be beneficial to talk more about ways to lessen this bias.

RE: We agree with the reviewer that the lack of blinding of participants or investigators increases the risk of performance bias. Unfortunately, the blinding was not intrinsically feasible in our trial. To lessen the bias we decided to use an objective outcome as primary outcome of our study (i.e. change of HbA1c 12 months after recruitment), and to involve experienced health professionals for the performance of the interventions, specifically sensitize and trained on good practices for the conduction of research studies. Indeed, we will take into account the potential implication of performance bias during the interpretation of our findings, including qualitative information arising from the co-evaluation phase of our study. We added a sentence in the manuscript to include details on this crucial point. We thank the reviewer for this comment. (see lines 275-281 page 13).

Reviewer #2: Authors plan to conduct a multicentre randomized controlled trial (RCT) to assess the effectiveness of a co-created health promotion intervention for immigrants with T2D. They will recruit 200 participants from four centers randomized into two arms: health promotion practice and experimental health promotion intervention. The focus will be on diet, physical activity, and therapeutic adherence.

1. Table 1 is unclear to this reviewer. Some more explanations for table 1 are needed.

RE: Ok, we tried to explain the table a little better. Lines 136-143 (page 5-6).

2. Data will be collected from four centers and it seems that the intervention will not be identical. If so, how to address the potential heterogeneity?

RE: We thank the reviewer for this comment. This is a crucial methodological issue when assessing co-created and context-adapted complex interventions in multicentre trials.

As presented in section 1.1.3 and Table 1, we decided to combine the standardization of the intervention by function (rather than by form) suggested by the Medical Research Council (Skivington K et al 2021) for the evaluation of complex interventions, with a careful conceptualization of what was the object of the evaluation (i.e. the contrast vs. the usual care defined by similar incremental effort). This led us to build an experimental design that could compare the novelty of the intervention in contrast to the existing usual care in centers with different backgrounds. Thus, the heterogeneity we should take into account will be the one emerging at function and incremental effort levels, which will be probably lower than the heterogeneity we would have taken into account considering the differences in the form of the intervention’s components. Despite this, whether the heterogeneity at function and incremental effort level will be high, proper analysis will be used to take it into account. We added a sentence on this point in the discussion section. (lines 354-356, page 16-17)

3. Line 263. Typo: enrolls not enrols

RE: we correct it.

4. Why won’t the statistician be blinded to group assignments? is there any reason to be unblinded?

RE: We thank the reviewer for this comment. Although the variable on the randomization arm can be masked, we consider it appropriate to consider the statistician unblinded because there are variables in the database describing the health promotion intervention that clearly differs between arms. These variables cannot be masked because they are among those to be included in the analysis.

5. What questionnaire will be used for interview?

RE: Whether the comment was referred to the questionnaires used to assess dietary habits and physical activity, we used the Mediterranean Diet Score, (MedDietScore) and the International Physical Activity Questionnaire - Short Form, IPAQ-SF, as reported in 2.1.9 paragraph on data collection.

Whether the comment was related to the questions included in the interviews performed during the co-creation process, no standardized questionnaires were used. Open-ended questions related to barriers and solutions were prepared ad-hoc by the qualitative research team and shared with the principal investigators of each recruiting center to guide the co-creation process. We included a sentence in the manuscript to clarify this point. (lines 212-213)

---

## [Decision Letter · Decision Letter 1]

8 Jan 2025

A randomized trial to evaluate a complex, co-created, culture-sensitive intervention to promote healthy lifestyles and compliance to therapy in immigrants with type 2 diabetes: a protocol of a multicenter Italian study

PONE-D-24-33957R1

Dear Dr. Francesco Venturelli

We’re pleased to inform you that your manuscript has been judged scientifically suitable for publication and will be formally accepted for publication once it meets all outstanding technical requirements.

Kind regards,

Omid Dadras, MD, PhD

Academic Editor

PLOS ONE

Additional Editor Comments (optional):

Reviewers' comments:

Reviewer's Responses to Questions

**Comments to the Author**

1. Does the manuscript provide a valid rationale for the proposed study, with clearly identified and justified research questions?

Reviewer #1: Yes

Reviewer #2: Yes

2. Is the protocol technically sound and planned in a manner that will lead to a meaningful outcome and allow testing the stated hypotheses?

Reviewer #1: Yes

Reviewer #2: Yes

3. Is the methodology feasible and described in sufficient detail to allow the work to be replicable?

Reviewer #1: Yes

Reviewer #2: Yes

4. Have the authors described where all data underlying the findings will be made available when the study is complete?

Reviewer #1: Yes

Reviewer #2: Yes

5. Is the manuscript presented in an intelligible fashion and written in standard English?

Reviewer #1: Yes

Reviewer #2: Yes

6. Review Comments to the Author

You may also provide optional suggestions and comments to authors that they might find helpful in planning their study.

Reviewer #1: Thank you for your detailed responses to the comments and for making the necessary revisions. I appreciate the clarifications and the improvements made to the manuscript, particularly in the area of the discussion of potential biases. The adjustments have significantly enhanced the clarity of the manuscript.

No further revisions are required at this stage.

Reviewer #2: Thanks for taking your time to address the comments. All raised comments have been satisfactory addressed.

7. PLOS authors have the option to publish the peer review history of their article (what does this mean? ). If published, this will include your full peer review and any attached files.

**Do you want your identity to be public for this peer review?** For information about this choice, including consent withdrawal, please see our Privacy Policy .

Reviewer #1: No

Reviewer #2: No

---

## [Editor Report · Acceptance letter]

PONE-D-24-33957R1

PLOS ONE

Dear Dr. Venturelli,

I'm pleased to inform you that your manuscript has been deemed suitable for publication in PLOS ONE. Congratulations! Your manuscript is now being handed over to our production team.

Kind regards,

on behalf of

Dr Omid Dadras

Academic Editor

PLOS ONE